# Regeneration of *Escherichia coli* Giant Protoplasts to Their Original Form

**DOI:** 10.3390/life9010024

**Published:** 2019-03-01

**Authors:** Kazuhito V. Tabata, Takao Sogo, Yoshiki Moriizumi, Hiroyuki Noji

**Affiliations:** Department of Applied Chemistry, The University of Tokyo, 7-3-1 Hongo, Bunkyo-ku, Tokyo 113-8656, Japan; sogo-takao@nojilab.t.u-tokyo.ac.jp (T.S.); yoshiki38_mn@nojilab.t.u-tokyo.ac.jp (Y.M.)

**Keywords:** giant protoplast, *Escherichia coli*, proliferation, cell wall-deficient bacteria

## Abstract

The spheroplasts and protoplasts of cell wall-deficient (CWD) bacteria are able to revert to their original cellular morphologies through the regeneration of their cell walls. However, whether this is true for giant protoplasts (GPs), which can be as large as 10 μm in diameter, is unknown. GPs can be prepared from various bacteria, including *Escherichia coli* and *Bacillus subtilis*, and also from fungi, through culture in the presence of inhibitors for cell wall synthesis or mitosis. In this report, we prepared GPs from *E. coli* and showed that they can return to rod-shaped bacterium, and that they are capable of colony formation. Microscopic investigation revealed that the regeneration process took place through a variety of morphological pathways. We also report the relationship between GP division and GP volume. Finally, we show that FtsZ is crucial for GP division. These results indicate that *E. coli* is a highly robust organism that can regenerate its original form from an irregular state, such as GP.

## 1. Introduction

Bacteria have the ability to survive under extreme environmental conditions, including high temperatures, high pressure, and intense UV radiation. They can also adapt well to extreme changes in the environment, such as the exposure to alcohol [1,2] or high temperature [3,4,5], or treatment with antibiotics [6,7]. More recently, the ability of bacteria to adapt to hybrid environments, such as their fusion to microchambers, has gained attention [8]. In this report, hybrid cells were prepared by replacing most of the cell membrane with inorganic materials (i.e., the microchamber wall). Interestingly, some of these hybrid cells (20% of total) had protein synthesis activity even after such a harsh treatment. In this way, bacteria have been shown to be robust organisms that can adapt to various environmental changes.

The cell wall, which is an important structure for bacterial survival, can be easily removed through treatment with enzymes or cell wall synthesis inhibitors [9,10]. Bacteria missing cell walls are called cell wall-deficient (CWD) bacteria [11,12], and are susceptible to osmotic lysis, but can be maintained by adjusting the external osmotic pressure [13]. CWD bacteria can be roughly categorized into spheroplasts, in which a part of the cell wall structure remains [14], and protoplasts, in which the cell wall is completely lost, leaving only the cell membrane [15,16]. Spheroplasts and protoplasts exhibit metabolic activity, but cannot undergo cell division due to the absence of a cell wall [17,18]. When placed in media, the bacteria regenerate to their original form prior to forming colonies [19,20,21]. In addition, the process of regeneration to the original bacteria by microscopic observation has also been obtained [22]. As described above, since CWD bacteria lose their cell wall mostly or entirely, they differ significantly from the original bacteria, but can regenerate to intact bacteria.

It has been reported that when cultured in an isotonic medium with a cell wall synthesis inhibitor, such as penicillin or cell division inhibitors, spheroplast or protoplast diameters increase from 1 to about 10 μm, resulting in giant protoplasts (GPs) [23,24,25]. Accordingly, the volume of GPs is much larger than the volume of the original bacteria. For example, the volume of *Escherichia coli* ranges from 0.2 to 3.0 fL depending on the nutritional condition of the culture [26,27,28,29,30]. Thus, assuming a GP of 10 µm, an *E. coli* volume of 1.5 fL would imply a GP volume of around 523 fL, or approximately 350 times larger than the original bacteria.

Observations of respiratory chain and anion channel activities have indicated that GPs possess a functional membrane [23,31]. In addition, DNA staining with DAPI or EtBr shows that GPs possess a large amount of DNA [31,32]. This is believed to be due to the replication of genomic DNA in GPs during the enlargement process. Thus, although GPs differ from the original *E. coli*, their membrane transport activity and DNA replication—which are important cell functions—are retained. From these reports, we concluded that GPs have a functional metabolism. However, whether they can regenerate into the original bacteria is unknown. The ability to return to their original state following the process of enlargement would add more evidence to the unexpectedly high robustness of bacteria as living organisms, and would provide more clarity on how bacteria survive extreme conditions.

In this paper, we prepared GPs from *E. coli* spheroplasts and examined whether protein synthesis in the GPs is functional. In addition, we cultured GPs in ampicillin-free medium and observed whether they could be regenerated to the original *E. coli* form. We also examined the volumes of GPs that can be regenerated by measuring their diameters. Finally, we report that FtsZ is involved in GP division during regeneration to the original *E. coli*. To conclude, we report that *E. coli* is a highly robust organism and that their GPs can be regenerated, despite having volumes several hundred times larger than the original wild-type bacteria.

## 2. Materials and Methods

### 2.1. Chemical Reagents and Bacterial Strains

Reagents were purchased from Wako unless otherwise stated. The microchamber used to observe giant protoplasts (GPs) was made from SU8-3050 (Nippon Kayaku). The GP culture medium consisted of 2.75% tryptic soy broth without dextrose (Difco Laboratories), 10 mM MgSO_4_, 300 mM sucrose, and 50 mM KCl. Antibiotics were added to the medium, as necessary. SP buffer (25 mM Tris-HCl (pH 7.4) and 400 mM sucrose) was used to form protoplasts in *E. coli*. BL21. *E. coli* BL21 was used except for in the FtsZ experiments. For green fluorescent protein (GFP) expression in GPs, GFPuv cloned into pET-9a (Novagen) was used, and its expression was performed by IPTG induction. *E. coli* expressing FtsZ-YFP (yellow fluorescent protein) was a kind gift from Dr. Masaki Osawa. FtsZ-YFP was expressed as reported previously [33].

### 2.2. Giant Protoplast Preparation

GPs were prepared as previously described [32]. Briefly, *E. coli* was cultured overnight and then placed in 10 mL LB broth until OD_660_ = 0.6. Following this, the mixture was centrifuged at 4000 rpm, 20 °C, for 5 min (MX-300, TOMY). Following centrifugation, the supernatant was discarded, and the pellet was resuspended in 10 mL SP buffer in a fresh test tube. To the test tube, we added 10 U/mL DNase I (Roche) and 400 µg/mL lysozyme, at final concentrations. The mixture was then incubated at 30 °C for 20 min. After incubation, the mixture was centrifuged at 4000 rpm, 20 °C, for 10 min, and the supernatant and pellet were separated. The pellet was suspended in 200 µL GP medium. The pellet was collected and suspended in 100 mL GP medium supplemented with 1 µg/mL ampicillin and 1 U DNase, and the mixture was cultured and shaken at 30 °C and 30 rpm.

### 2.3. Recovery from GP

To allow GPs to revert to the original *E. coli* form, GPs cultured in the GP medium were centrifuged (4000 rpm, 20 °C, 10 min), the supernatant was discarded, and the pellet was placed in GP medium without ampicillin. The mixture was placed into a microchamber and observed under a microscope at 30 °C.

### 2.4. Microchamber Array Fabrication

To allow for the extended observation of GP morphology, a microchamber 50 µm in diameter and 50 µm in height was prepared. A cover glass 30 mm in diameter (No. 1 Matsunami) was washed and spin-coated with SU8-3050 (500 rpm for 10 s followed by 2500 rpm for 50 s). The glass was then baked for 20 min at 65 °C and 12 min at 95 °C. After being allowed to slowly return to room temperature, the fabricated component was exposed to BA160 mask aligner (Nanometric Technology Inc.) and developed with SU8 developer. Finally, the component was adhered to a 35 mm dish in which a 27 mm-wide hole was drilled into the bottom surface. 

### 2.5. Microscopic Observations and Analysis

Images were taken with a CMOS camera (ORCA-Flash4.0, Hamamatsu) attached to an inverted microscope (Ti-E, Nikon). Image analysis of the GP morphology was performed using NIS-Elements AR (Nikon). We measured the diameter from the image of the microscope and calculated the volume of the GP as a sphere. When the shape changed, the long axis and the short axis were measured to approximate the shape as a cylinder or a sphere, and the volume was calculated by considering both. The fluorescence intensity of GFP was measured as the average intensity by enclosing a single GP with ROI from the fluorescence microscopic image.

## 3. Results

### 3.1. Giant Protoplast (GP) Size and Incubation Time

GPs are known to reach 10–30 μm in diameter after cultivation for 24 h. We examined the relationship between incubation time and diameter in more detail (Figure 1a,b). Culture was performed using GP medium plus ampicillin, and protoplast diameters were measured by imaging analysis. The blue dot in Figure 1a shows the diameter immediately following lysozyme treatment of *E. coli*. We marked t = 0 h as the moment we switched the medium to the GP medium minus ampicillin. The diameter of the spheroplasts, which was approximately 3 μm at 0 h, increased with the culture time, reaching an average of 11 μm at 8 h. After 8 h, we found no evidence of diameter change, indicating that the GPs had reached their maximum size by this time. Only a small number of GPs reached 20–30 μm in size. We also investigated the relationship between culture time and volume (Figure 1a, inset). For the first 6 h of culture, the relationship was found to depend on the replication time, and could be modeled with the following equation:(1)V=Vint×2tτ.
Here, *V* represents the volume of the GP, *V_int_* represents the initial volume, *t* is time, and *τ* represents the time to double the volume. As a result of this analysis, we found that *τ* was 1.12 h. 

Next, protein expression in GPs was examined. GPs were transfected with a plasmid encoding green fluorescent protein (GFP) that can be induced with IPTG. A histogram of GFP fluorescence at different culture times is shown in Figure 1c. The distribution of the histogram shifts to higher intensities up to 8 h, indicating that the GFP expression level had increased. However, the distribution shifted to lower intensities at 22 h. These results suggest that protein synthesis activity increased up to 8 h from the start of the culture, but diminished by 22 h. These observations are consistent with observations of changes in the GP diameter and volume shown in Figure 1a. In other words, GP diameters increase while protein synthesis is occurring.

### 3.2. Regeneration of GPs

The diameter and volume of GPs were about 10 and 1000 times larger, respectively, than those of *E. coli* spheroplasts. We examined whether GPs can undergo cell division and regenerate into *E. coli*. Figure 2a provides a schematic of the experiment. First, *E. coli* was cultured to OD_600_ = 0.6 and treated with lysozyme. Then, the spheroplasts were added to GP medium plus ampicillin and cultured at 30 °C. After culturing, the cells were harvested by centrifugation and suspended in GP medium minus ampicillin. This GP suspension was subsequently introduced into a microchamber array (φ = 50 μm, h = 100 μm) prepared for microscopic observation. After 1.5–2 h, we began time lapse observation at 30 °C. After 1 h, we found GPs expressing two elongated poles (an example is shown in Figure 2b). After 5 h, structures resembling septa were seen. The GPs continued to elongate and division was observed repeatedly. Among our observations of 1441 GPs, 73.6% remained spherical, 20.4% showed deformation of the spherical shape but did not divide, and the remaining 5.9% showed deformation of the spherical shape and did divide (Figure 2c). Among the latter group, 57.6% divided by expressing the bacilliform and elongating into a rod shape, 30.6% divided from an amorphous state, and the remaining 11.8% formed three or more poles, and elongated and divided from each pole (Appendix A). Overall, we confirmed that GPs can regenerate and divide.

Plots of the volume distribution of dividing GPs are presented in Figure 3a. The volume at 0 h was widely distributed, from approximately 50–2500 fL with a mean of 546 fL (Figure 3a, red), but upon regeneration ranged primarily between 100 and 350 fL with a mean of 237 fL (Figure 3a, blue). In GPs of 350 fL or more, cell division was drastically decreased. Furthermore, we found that GPs lost their regenerative ability at over 350 fL (about 4 h of culturing). 

We then examined whether regenerated GPs could form colonies on agar plates. GPs whose ability to divide once or twice had been confirmed by microscopy were collected from the microchamber with a microglass pipette and spotted on an agar plate (Figure 3b). In total, 20 GPs were spotted, and six colonies were obtained. By contrast, with the negative control, in which the contents of the microchambers with no GPs were spotted on the agar plates, no colonies were seen across a total of 16 experiments. We measured and compared the volume of GPs during cell division with that of wild-type (WT) *E. coli* (Appendix A). In this experiment, focusing on the dividing GPs, the volume was measured over time (0, 4, and 8 h) and plotted in the histogram. At 0 h, 10 GPs had increased to 49 cells at 8 h. The cell volume was estimated as a sphere or a cylinder by measuring the respective cell sizes of the GPs and *E. coli*. We found the volume of the dividing GPs diminished with time, and that the new cells had a volume consistent with the WT *E. coli* volume over the first 8 h. Furthermore, microscopic images showed that the divided cells were rod-shaped, like the WT *E. coli*. Thus, a single GP could form a colony by regenerating and dividing, albeit at low efficiency. Overall, these results show that GPs can regenerate into the original *E. coli* form, as determined from their shape and colony-forming ability.

### 3.3. Deformation and Division of GPs

In *E. coli*, division is dependent on FtsZ, a filament protein that forms a contracted ring. During the regeneration process, GPs greatly change their shape, forming poles. In order to investigate whether FtsZ is involved in this process, we prepared FtsZ-YFP and observed its localization in GPs (Figure 4). In the original *E. coli* cells, yellow fluorescent protein (YFP) fluorescence was observed as bands. In spherical GPs, no localized YFP fluorescence was observed. However, in deformed GPs, a ring-like distribution was observed in the deformed portion (arrow in Figure 4a). Bands could also be seen in the constricted region. We also observed localization of FtsZ-YFP during the GP regeneration process. The arrows show that FtsZ-YFP aggregated in the constricted part of GPs, which was the location at which GPs divided. These observations suggest that FtsZ is involved in the deformation and division of GPs.

## 4. Discussion

Although the division and proliferation of various CWD bacteria have been reported, this is the first study to show that *E. coli* GPs can regenerate, divide, and proliferate. Culturing *E. coli* spheroplasts in an isotonic medium resulted in GPs that reached an average diameter of about 10 μm in about 8 h. The cell volume doubling time was calculated from the relationship between volume and division to be approximately 1.12 h. In comparison, the doubling time of *E. coli* in a rich medium is known to be about 20 min, but varies greatly depending on the culture conditions. Moreover, Schmidt et al. reported that the doubling time of *E. coli* varies from 18 to 162 min depending on the nutritional conditions [34]. Therefore, the doubling of GP volumes in 1.12 h is comparable to the growth rate of WT *E. coli*. Thus, the spheroplasts, which were 1.1 µm in diameter and 0.75 fL in volume after lysozyme treatment, doubled their volumes about 10 times before reaching their maximum size. Further, we found the GFP fluorescence intensity of GPs increased up to 8 h, but tended to decrease thereafter, suggesting that the increase in GP size correlated with protein expression. Previous studies using GPs have performed GP culture for about 24 h [31,32], but our results suggest that 6 to 8 h of GP culture is sufficient.

We attempted to regenerate *E. coli* by culturing GPs in a medium without ampicillin. Under these growth conditions, a small percentage (5.9%) of GPs regenerated and divided, and displayed highly similar morphologies to the original *E. coli*. Among these GPs, we observed many irregular shapes during division, as well as the formation of three or more poles. The regeneration process showed a great deal of similarity to that of *B**acillus megaterium*, a Gram-positive bacterium [22], and *Geotrichum candidum*, a type of fungus [20]. Identical pole formation has been observed in the protoplasts of *E. coli* [25], suggesting that GPs and protoplasts have similar elongation processes. On the other hand, unlike *Bacillus*, another Gram-positive bacterium [35], GPs did not display dumbbell shapes during division, most likely because of differences in the outer layer structure of cell membrane of Gram-negative and Gram-positive bacteria.

After confirming their division under a microscope, we collected 20 GPs from individual microchambers using a microglass pipette and spotted them onto agar plates. Of the 20, 6 formed colonies and reverted to the original *E. coli* form. The low efficiency of colony formation could be a technical limitation of the experimental system resulting from the GPs not being discharged from the microglass pipette in an effective manner. Furthermore, the colonies were all fried egg-like colonies, which is more consistent with L-form bacteria than with *E. coli* (Figure 3b) [36]. L-form bacteria are a well-known variety of CWD bacteria which can be prepared from *E. coli* and *B. subtilis* in the presence of cell wall synthesis inhibitors or genetic modification [37,38,39]. L-form cell division is characterized by an extrusion-resolution process in which a part of the membrane fraction protrudes and divides [40]. Therefore, L-form bacteria do not require a mitotic apparatus like FtsZ for division. Since L-form bacteria can only divide at the cell membrane, they are believed to reflect the division and proliferation mode of primitive cells, and have drawn attention from the perspective of the origins of life [41,42]. Microscopic observation of the colonies from Figure 3b resuspended revealed a mixture of spherical cells and rod-shaped cells (Appendix A). Considering the fried egg-like colonies, we believe that spherical cells are unstable L-form bacteria. In *E. coli*, the probability of obtaining an L-form in a medium supplemented with penicillin is less than 0.1% [25]. The extraordinarily high rate of fried egg-like colonies in our experiments may therefore be the result a hypertonic culture medium in the agar plate and extended exposure to ampicillin.

Based on the investigation of GPs over 4–8 h of culture, we found that regeneration occurred at an average cell volume of 237 fL. The L-form of *E. coli* which, similar to protoplasts, does not have an outer membrane, has a diameter of 1.33 ± 0.50 μm, which implies a volume of 1.23 fL [43]. We found the volume of spheroplasts following lysozyme treatment to be 0.75 fL (Figure 1a, blue dot), while that of intact *E. coli* was 0.2–3.0 fL. Thus, the volume of dividing GPs was between 79 and 1186 times greater than that of *E. coli*. Interestingly, GPs exceeding 350 fL, which corresponded to a diameter of approximately 7.5 μm (Figure 1a), were rarely seen to regenerate. From Figure 4, because the FtsZ protein is believed to play an essential role in GP deformation and division, it is possible that the system responsible for the division of *E. coli* has an upper limit in which it can deform and divide the cell. On the other hand, Leaver et al. showed that FtsZ is not required for the division of L-form bacteria, such as CWD bacteria like GPs [40]. Moreover, *E. coli* with cell walls can also divide without FtsZ [44]. However, this phenomenon was only seen in mutant *E. coli* (*∆lpp* and *∆lpoB* or *∆mrcB* gene) and not the wild-type BL21 strain we used in our experiments. Furthermore, the GPs in our experiments are not L-form. Therefore, our observations in Figure 4 suggest that FtsZ is required for the deforming and division of GPs.

In conclusion, in this study, we demonstrated that *E. coli* is able to regenerate, divide, and proliferate following changes in cell volume more than 100 times greater than the original wild-type state. These results indicate that *E. coli* is a highly robust organism.

## Figures and Tables

**Figure 1 life-09-00024-f001:**
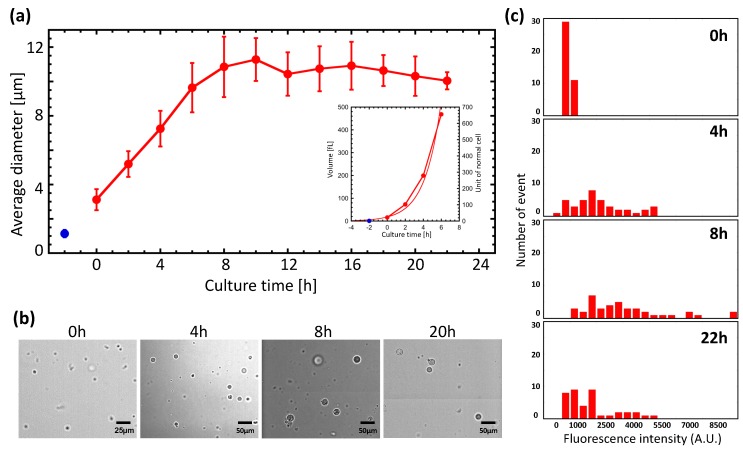
Increased diameter and protein expression with a giant protoplast (GP) culture. (**a**) Plot of GP diameter versus incubation time. Red dots represent average GP diameters. The blue dot is the average diameter of *Escherichia coli* immediately after lysozyme treatment. The start of microscopic observation of the sample is indicated by 0 h. (**b**) Photographs of GP at different incubation times. (**c**) Histogram of GFP fluorescence intensity at different incubation times. At each culture time, IPTG was added to induce GFP expression, and after 2 h the fluorescence intensity was measured.

**Figure 2 life-09-00024-f002:**
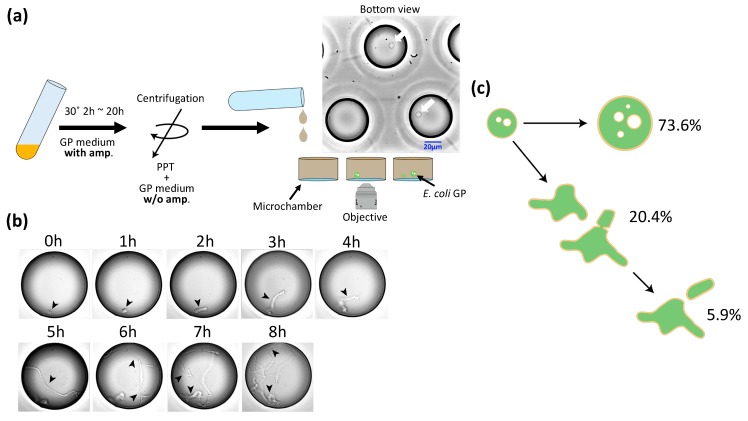
Observation of the GP regeneration process. (**a**) Schematic for the observation of GP regeneration. After culturing, the medium was changed to a medium without ampicillin (amp) by centrifugation, and GPs were then introduced into the microchamber. The photograph shows an actual image. White arrows show GPs. (**b**) Time lapse imaging of the GP regeneration process. The diameter of the microchamber is 50 μm. Arrows indicate regenerating bacteria. (**c**) Percentage of GPs whose morphology changed in the regeneration experiment, where 73.6% did not change, 20.4% deformed, and 5.9% showed cell division.

**Figure 3 life-09-00024-f003:**
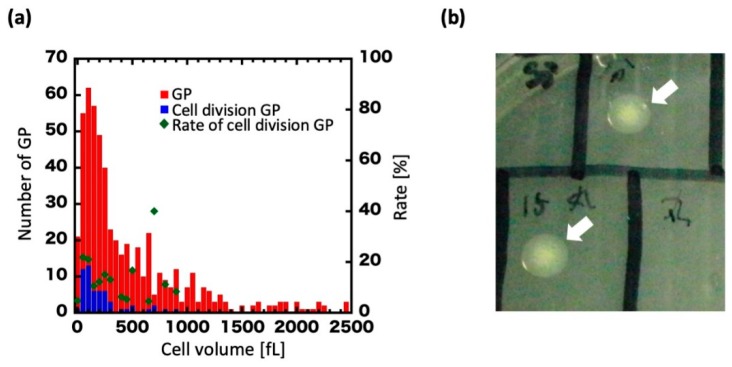
Cell volume of regenerating GPs and colony formation. (**a**) Histogram of GP volume at 0 h. Blue dots show the volume of GPs that underwent cell division. Red dots show the volume of all GPs observed. Green dots show the percentage of GPs that divided at a given volume, which is defined as the ratio of blue dots to red dots (right vertical axis). (**b**) A colony formed on an agar plate (arrows).

**Figure 4 life-09-00024-f004:**
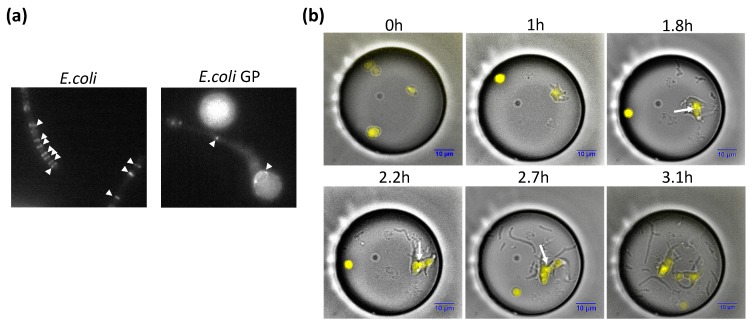
Localization of FtsZ in GP. (**a**) Localization of FtsZ-YFP (yellow fluorescent protein) in GP and wild-type (WT) *E. coli*. (**b**) Cell division of GP during GP regeneration and localization of FtsZ-YFP. Merged bright field images and fluorescent images of the GP regeneration process are shown. White arrows indicate FtsZ-YFP localized at the division interface.

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
