# Peer review of "Regeneration of Escherichia coli Giant Protoplasts to Their Original Form"

_life, 2019, doi:10.3390/life9010024_

Round 1

Reviewer 1 Report

The paper by Tabata et al. showed the regeneration of E. coli GP. Detailed characterization of the GP was performed mostly through microscopic observation. The experiments were performed and presented properly. I only have minor comments as listed below.

Line 42: “but can maintained” -> “but can be maintained”

Line 49: “microscopic observation has been observed” -> “microscopic observation has been obtained”

Line 51, “are” is not needed

Line 59-60: It is not clear where the 523 fL came from.

Line 67: There is something wrong with “ but not whether it can regenerate into the original bacteria”. I can’t get the meaning.

Line 136: How did the authors quantify the volume of the GP? In particular, if the GP is not spherical it shouldn’t be easy to quantify the volume. Please describe.

Line 145: How is the GFP fluorescence quantified? Whole cell intensity or concentration?

Line 148: “Protein synthesis” -> “protein synthesis activity”

Line 170: no need “and”

Line 224: Arrow is not present in Fig 4a.

Line 227: Authors claim that ring shape is present, presented by the arrows in Fig. 4b. I do agree in seeing the ring structure in Fig. 4a E.coli, but not in b. It just looks like small aggregates. Can the authors describe the aggregates in a better way?

Line 253: Why all of a sudden discussing about the patch clamp experiments? It is not easy to understand.

Line 384: e.coli  should be E. coli

Author Response

We thank the reviewers for their comments. Below we write those comments in blue italicsand our responses in black.

We greatly appreciate the comments and careful reading by Reviewer 1 that helped improve the manuscript.

Line 42: “but can maintained” -> “but can be maintained”

Line 49: “microscopic observation has been observed” -> “microscopic observation has been obtained”

Line 51, “are” is not needed

Line 148: “Protein synthesis” -> “protein synthesis activity”

Line 170: no need “and”

Line 224: Arrow is not present in Fig 4a.

Line 384: e.coli  should be E. coli

We apologize for the errors and thank the reviewer for the careful reading of our manuscript. We have revised the manuscript accordingly ( line 42, 49, 51, 154, 177, Fig.4a and 397).

Line 59-60: It is not clear where the 523 fL came from.

The volume of the GP comes from assuming a diameter of 10 µm. With this assumption,an E. colivolume of 1.5 fL would imply a GP volume of around 523 fL, which is approximately 350 times larger the original bacteria. We have revised the manuscript to make this point clear (line 59).

Line 67: There is something wrong with “ but not whether it can regenerate into the original bacteria”. I can’t get the meaning.

We have rewritten the statement in the revised manuscript as follows.

From these reports, we concluded that GP has a functional metabolism.However,whetherit can regenerate into the original bacteria is unknown (line 66-68).

Line 136: How did the authors quantify the volume of the GP? In particular, if the GP is not spherical it shouldn’t be easy to quantify the volume. Please describe.

We measured the diameter from the microscopic image and calculated the volume of the GP as a sphere. When the shape changed, the long axis and the short axis were measured to approximate the shape as a cylinder or a sphere, and the volume was calculated by combining them. We have revised the manuscript to reflect this point (line 125-129).

Line 145: How is the GFP fluorescence quantified? Whole cell intensity or concentration?

The fluorescence intensity of GFP was measured as the average intensity by enclosing a single GP with an ROI in the fluorescence microscopic image.We have revised the manuscript to reflect this point (line 129-130).

Line 227: Authors claim that ring shape is present, presented by the arrows in Fig. 4b. I do agree in seeing the ring structure in Fig. 4a E.coli, but not in b. It just looks like small aggregates. Can the authors describe the aggregates in a better way?

Along this suggestion, we revised the manuscript in this point. 

The arrows show that FtsZ-YFP aggregated in the constricted part of GP, which was the location at which GP divided (line 233-234).

Line 253: Why all of a sudden discussing about the patch clamp experiments? It is not easy to understand.

We agree with the reviewer and have changed the text as follows.

Previous studies using GP have performed GP cultivation for about 24 hours, but our results suggest that 6 to 8 hours of GP culture is sufficient (line 258-260).

Reviewer 2 Report

Comments to authors

The authors showed in this report that GP derived from E. coli can revert to the original morphology of E. coli under microscope and was able to form colonies on plate in the absence of ampicillin. I think that experiments were well done. However, I have some comments on the manuscript.

1. Qualities of pictures are very low

Fig. 1b, 2b, and 4: contrasts of the images are low.

Fig. 4: bright images of cells are hardly seen.

2. I recommend the authors to show the magnified images of a cell with their wide view of the images. Especially, I can’t see GP cell in Fig.2b 0h and I can’t recognize a ring like shape of FtsZ-YFP in Fig. 4b.

3. Fig. 1c: What kind of the expression system was used for the expression of GFP? Did they use a plasmid to express GFP (vector, antibiotics?) or did they integrate GFP gene in E. coli genome?

4. The authors concluded that FtsZ is crucial for GP division because FtsZ-YFP is localized to the constricted part of GP cells. However, the localization of FtsZ to the constriction site does not mean that FtsZ is crucial for GP division. If the authors obtained a result that an ftsZ depleted strain did not show deformation and division of GP cells, then they can conclude that FtsZ is crucial for GP division. In fact, Mercier et al., (2016) showed that FtsZ is not required for recovery to rod shape from B. subtilis L-form. The authors should cite and discuss the paper.    

Author Response

We thank the reviewers for their comments. Below we write those comments in blue italicsand our responses in black.

We greatly appreciate the comments and careful reading by Reviewer 2 that helped improve the manuscript.

1. Qualities of pictures are very low

Fig. 1b, 2b, and 4: contrasts of the images are low.

Fig. 4: bright images of cells are hardly seen.

We apologize for the inconvenience and appreciate the reviewer 2 for kindly informing us of such qualities of pictures. We adjusted the contrast of the picture.

2. I recommend the authors to show the magnified images of a cell with their wide view of the images. Especially, I can’t see GP cell in Fig.2b 0h and I can’t recognize a ring like shape of FtsZ-YFP in Fig. 4b.

Such as comment 1, we made a contrast adjustment. Also, we highlighted the GP in Fig. 2b with arrows. Regarding Fig.4b, Reviewer 1 made a similar comment. We therefore explain in the revised manuscript that the image was used to identify aggregates.

3. Fig. 1c: What kind of the expression system was used for the expression of GFP? Did they use a plasmid to express GFP (vector, antibiotics?) or did they integrate GFP gene in E. coli genome?

To answer these questions, we have added the following to the revised manuscript (line 87-91).

E. coli BL21 was used except for FtsZ experiments. For the GFP expression in GP, GFPuv cloned into pET-9a (Novagen) was used, and its expression was performed by IPTG induction. E. coliexpressing FtsZ-YFP was a kind gift from Dr. Masaki Osawa.FtsZ-YFP was expressed as reported previously [33].

4. The authors concluded that FtsZ is crucial for GP division because FtsZ-YFP is localized to the constricted part of GP cells. However, the localization of FtsZ to the constriction site does not mean that FtsZ is crucial for GP division. If the authors obtained a result that an ftsZ depleted strain did not show deformation and division of GP cells, then they can conclude that FtsZ is crucial for GP division. In fact, Mercier et al., (2016) showed that FtsZ is not required for recovery to rod shape from B. subtilis L-form. The authors should cite and discuss the paper. 

We thank the reviewer for kindly letting us know the relevant work. We cite this paper in the discussion part. Along the suggestion, we added following text (line 306-312).

On the other hand, Leaver et al. showed that FtsZ is not required for the division of L-form bacteria such as CWD bacteria like GP [40]. Moreover, E. coli with cell walls can also divide without FtsZ [44]. However, this phenomenon was only seen in mutant E. coli ( ∆lppand ∆lpoBor ∆mrcB gene ) and not the wild type BL21 strain we used in our experiments. Furthermore, the GP in our experiments are not L-form. Therefore, our observations in figure 4 suggests that FtsZ is required for the deforming and division of GP.